# Development of Oxygen-Bridged Pyrazole-Based Structures as Cannabinoid Receptor 1 Ligands

**DOI:** 10.3390/molecules24091656

**Published:** 2019-04-27

**Authors:** Gabriele Murineddu, Battistina Asproni, Paola Corona, Sandra Piras, Paolo Lazzari, Stefania Ruiu, Laura Legnani, Lucio Toma, Gérard A. Pinna

**Affiliations:** 1Department of Chemistry and Pharmacy, University of Sassari, Via F. Muroni 23/A, 07100 Sassari, Italy; asproni@uniss.it (B.A.); pcorona@uniss.it (P.C.); piras@uniss.it (S.P.); pinger@uniss.it (G.A.P.); 2KemoTech SrL, Build 3, Loc. Piscinamanna, 09010 Pula, Italy; paolo.lazzari@kemotech.it; 3National Research Council (C.N.R.)-Institute of Translational Pharmacology, U.O.S. of Cagliari, Science and Technology Park of Sardinia Polaris, 09010 Pula, Italy; stefania.ruiu@ift.cnr.it; 4Department of Chemical Sciences, University of Catania, Viale A. Doria 6, 95125 Catania, Italy; laura.legnani@unict.it; 5Department of Chemistry, University of Pavia, Via Taramelli 12, 27100 Pavia, Italy; lucio.toma@unipv.it

**Keywords:** cannabinoid receptor 1 neutral antagonists, synthesis, structure-activity relationship studies, food intake, gastrointestinal transit, modelling studies

## Abstract

In this work, the synthesis of the cannabinoid receptor 1 neutral antagonists 8-chloro-1-(2,4-dichlorophenyl)-*N*-piperidin-1-yl-4,5-dihydrobenzo-1*H*-6-oxa-cyclohepta[1,2-*c*]pyrazole-3-carboxamide **1a** and its deaza *N-*cyclohexyl analogue **1b** has led to a deepening of the structure-activity studies of this class of compounds. A series of novel 4,5-dihydrobenzo-oxa-cycloheptapyrazoles analogues of **1a**,**b**, derivatives **1c**–**j**, was synthesized, and their affinity towards cannabinoid receptors was determined. Representative terms were evaluated using in vitro tests and isolated organ assays. Among the derivatives, **1d** and **1e** resulted in the most potent CB_1_ receptor ligands (K*_i_*CB_1_ = 35 nM and 21.70 nM, respectively). Interestingly, both in vitro tests and isolated organ assays evidenced CB_1_ antagonist activity for the majority of the new compounds, excluding compound **1e**, which showed a CB_1_ partial agonist behaviour. CB_1_ antagonist activity of **1b** was further confirmed by a mouse gastrointestinal transit assay. Significant activity of the new CB_1_ antagonists towards food intake was showed by preliminary acute assays, evidencing the potentiality of these new derivatives in the treatment of obesity.

## 1. Introduction

Cannabinoid receptors [1], CB_1_R [2] and CB_2_R [3], are part of the endocannabinoid system (ECS). This system consists of cannabinoid receptors, endogenous ligands, and several proteins responsible for their synthesis and degradation. Cannabinergic neurotransmission is closely associated with cognition, reward, appetite, pain perception and neuroexcitability, to name just a few putative physiological functions. Thus, it appears that dysfunction of the ECS contributes to several pathophysiological conditions that have been associated with the aforementioned biological processes [4].

Pharmacologic interest in CB_1_ antagonist derivatives is particularly related to their established effects on food intake and weight control, improvement of cardiovascular risk factors, smoking cessation and treatment of alcohol and drug abuse [5], liver fibrosis [6], and erectile dysfunction [7]. Among synthetic cannabinoid compounds, the CB_1_ inverse agonist pyrazole derivative rimonabant (SR14176A) produced by Sanofi-Aventis reached the market in Europe (as Acompliat^®^) for the treatment of obesity and related metabolic risk factors. Unfortunately, the European Medicines Agency blocked its sales due to evidence of related adverse effects, in particular increased incidence of depression and suicides [8]. This event encouraged pharmaceutical companies to block or reduce investments in cannabinoid-based drug discovery. However, an imposing body of studies is in progress which considers the significant potentiality of this class of substances. Particular attention is focused on the individuation of new CB_1_ antagonist compounds which acts like rimonabant on the reduction of appetite and weight control but has reduced or neglected side effects, i.e., by acting at the peripheral level without having an effect on the central nervous system [8].

Recently we have reported two neutral CB_1_R antagonists, compound **1a** [9] and its deaza analogue **1b** [10], as seen in Figure 1, demonstrating that the chronic treatment of obese mice with **1a**, endowed with poor blood-brain barrier (BBB) permeability, induces the reduction of food intake and body weight without inducing the side effects responsible for withdrawal of rimonabant, whereas **1b** reduces the food intake in rats through a dopamine-dependent mechanism, suggesting an action on the hedonic component of food intake.

In general, it has been highlighted that chronic treatment with compound **1a** results in both anti-obesity effects and cardiovascular risk factor improvement in C57BL/6N diet-induced obesity (DIO) mice and fat diet (FD) mice [9]. Moreover, in contrast to SR141716A treatment, the chronic administration of **1a** does not change mRNA expression of both monoaminergic transporters and neurotrophins, which are highly related to anxiety and mood disorders. These previously reported results [9] suggest that compound **1a** reduces body weight and can restore the disrupted expression profile of genes linked to the hunger-satiety circuit without altering monoaminergic transmission, probably avoiding SR141716A side effects.

Food intake, food self-administration, and electrophysiological studies were carried on the deaza-analogue of **1a**, compound **1b**. Behavioural studies evidenced that at the i.p. dose of 0.125 mg/kg, **1b** reduced both food intake and body weight. Furthermore, the i.v. administration of high doses of **1b** reverted the effect of the agonist WIN-55,212-2 on the activity of ventral tegmental area (VTA) dopamine neurons projecting to the NAc shell, confirming the CB_1_ antagonism of these novel 4,5-dihydrobenzo-oxa-cycloheptapyrazoles [10].

Hence, the novel CB_1_ neutral antagonists **1a** and **1b** may represent useful candidate agents for the treatment of obesity and its metabolic complications.

Encouraged by these results, our efforts have been aimed towards the design and synthesis of new CB_1_R neutral antagonists unable to cross the BBB, to avoid adverse effects determined by the use of SR141716A. Thus, we assumed compound **1a** as the lead compound for the further characterization of the structure-activity relationship studies within the **1a** class of compounds, with an aim to disclose new potential substitutes of SR141716A with reduced side effects. Moreover, we wanted to further confirm the CB_1_ antagonism effect of these templates by employing the gastrointestinal test. We report the preparation, cannabinoid affinity evaluation and in vitro, in ex-vivo and in vivo assessment of novel 4,5-dihydrobenzo-1*H*-6-oxa-cyclohepta[1,2-*c*]pyrazoles (**1c**–**j**) bearing different substituents in position 3 of the pyrazole ring (Table 1).

## 2. Results and Discussion

### 2.1. Chemistry

The six-step synthetic pathway for the preparation of the 8-chloro-1-(2,4-dichlorophenyl)-4,5-dihydrobenzo-1*H*-6-oxa-cyclohepta[1,2*-c*]pyrazole-3-carboxamides **1a**–**j** is outlined in Scheme 1.

The reaction between *m*-chloro-phenol (**2**) and gamma-butyrrolactone gave the acid **3**, which upon cyclization with polyphosphoric acid (PPA) afforded the ketone **4** [11]. The readily available ketone was converted to the hydroxyketoester **5** using a Claisen reaction with diethyl oxalate in the presence of sodium ethylate. The ketoester **5** was then treated with 2,4-dichlorophenylhydrazine hydrochloride and refluxing in ethanol to furnish the pyrazole **6**. Intermediate **6** was hydrolysed to give the corresponding acid **7**. The acid **7** reacted with 1-ethyl-3-(3-dimethylaminopropyl)carbodiimide hydrochloride (EDC), 1-hydroxybenzotriazole (HOBt) and the appropriate amine or hydrazine to afford the desired derivatives **1a**–**j**.

### 2.2. Biology

Affinities at CB_1_R and CB_2_R for compounds **1c**–**j** were assessed by competition of [^3^H]-CP-55,940 in mouse brain (minus cerebellum) and spleen homogenates, respectively. Radioligand binding experimental procedures previously reported by us [12] were adopted. The results were compared with the *K_i_* values of the leads and reference compound, **1a**, **1b** and SR141716A, respectively.

Agonist or antagonist activity of the new derivatives towards CB_1_Rs were assayed through in vitro testing based on the determination of phosphorylated ERK 1/2 (P-ERK 1/2) expression in a mouse neuroblastoma N1E-115 cell line. As previously reported [13], this cell line expresses CB_1_Rs but not CB_2_Rs; moreover, it has been shown that the exposure of N1E-115 cells to the cannabinoid agonists arachidonyl-2-chloroethylamide (ACEA) and WIN 55-212,2 induces a rapid phosphorylation and activation of the ERK 1/2. The addition of CB_1_ antagonists as SR141716A abolishes the response induced by the cannabinoid agonists.

For all of the new derivatives which had good CB_1_ affinity (*K_i_* lower than 60 nM), the in vitro test results were successively confirmed and examined carefully by both isolated organ assays (mouse vas deferens) and in vivo capability to affect gastrointestinal motility. The mouse isolated vas deferens is a nerve-smooth muscle preparation that serves as a highly sensitive and quantitative functional in vitro bioassay for cannabinoid CB_1_R agonists [14]. The effect of CB_1_ agonists is related to the action on naturally expressed prejunctional neuronal CB_1_Rs to inhibit release of the contractile neurotransmitters, noradrenaline and ATP, which is provoked by electrical stimulation [15]. It has been shown that CB_1_ antagonist derivatives are able to block or reduce the inhibition of the electrically evoked contraction amplitude of the vas deferens induced by CB_1_ agonists, with parallel dextral shifts in CB_1_R agonist log concentration-response curves in electrically stimulated tissues [16]. According to previous studies concerning cannabinoid agents, a gastrointestinal transit test was adopted to confirm CB_1_ antagonism [17]. In fact, CB_1_Rs are significantly expressed in the enteral system of various mammalians and the gastrointestinal motility is modulated by their action [18]. Moreover, the capability of CB_1_ antagonist compounds both to block the constipation effect induced by cannabinoid agonists and to themselves increase themselves gastrointestinal motility has been reported [19].

To further evaluate the potential application of these CB_1_ antagonists in the control of food intake and in obesity treatment, compound **1b** was preliminarily tested in comparison with SR141716A in an acute in vivo assay using C57Bl6/J mice. The evaluation was carried out according to the animal model previously reported by Di Marzo [19].

#### 2.2.1. Radioligand Binding Assays

Investigation of the cannabinoid receptors affinities of the 8-chloro-1-(2,4-dichlorophenyl)*-*4,5-dihydrobenzo-1*H*-6-oxa-cyclohepta[1,2-*c*]pyrazole core was focused on the carboxamide moiety by introduction of different types of substituents. Two main subclasses of the derivatives were investigated and are reported in Table 1: compounds **1c**–**e** and **1f**–**j** bear saturated heterocyclic amines and myrtanyl moieties or aryl substituents in position 3 of the pyrazole ring, respectively.

The *N*-piperidinyl carboxamide **1a** [9], with a CB_1_-affinity *K_i_* value of 10.25 nM, was assumed as the lead compound for the pharmacomodulation of this new tricyclic system. First, we wanted to assess the effect of the homologation of the piperidine ring of **1a**, synthesizing the pyrrolidine (**1c**) [20] and homopiperidine (**1d**) analogues. This simple variation of a methylene unit in the aza-cycloalkyl substituent in position 3 of the pyrazole ring resulted in a 5.4-fold and 3.5-fold CB_1_-affinity decrease, respectively. To further evaluate the effect of an alicyclic moiety in the carbamoyl group on the tricyclic core, we designed the replacement of the cyclohexyl ring of **1b** [10] with the introduction of a myrtanyl moiety (**1e**). Among the reported derivatives, all of the heterocycloalkyl substituted compounds (**1c,d**) showed both good affinity (*K_i_*CB_1_ < 60 nM) and significant selectivity (*K_i_*CB_2_/*K_i_*CB_1_ > 57) toward CB_1_Rs, whereas the myrtanyl derivative (**1e**) allowed the maintaining of CB_1_ affinity at the same level of the derivatives **1a**–**d**, even if with a loss of CB_1_R versus CB_2_R selectivity.

To examine the influence of the variously substituted benzene ring in position 3 of the pyrazole ring on the CB_1_R affinity of the new tricyclic template, compounds **1f**–**j** were synthesized. The simple replacement of the piperidine group with the phenyl ring (**1f**) in the carboxamide moiety determined a 23-fold CB_1_-affinity decrease (*K_i_*CB_1_ = 233 nM). This trend was observed for all other aryl carboxamides. In particular, the introduction of a –Cl atom (**1g**) in the para position of the phenyl ring or a –OCH_3_ group (**1i**) produced a CB_1_ affinity decrease compared to **1f**. In addition, the para*-*substitution with a –CH_3_ (**1h**) or a –CF_3_ group (**1j**) resulted in a marked unfavourable effect on the CB_1_ affinity. All tested compounds showed a good or high CB_1_ versus CB_2_ selectivity.

#### 2.2.2. In Vitro Assays

Contrary to WIN 55,212-2, no effect on the P-ERK 1/2 expression was detected when mouse neuroblastoma N1E-115 cells were treated with compounds **1b**–**d** at concentrations up to 1 μM, evidencing no agonist activity of these derivatives towards the CB_1_ receptor. On the contrary, according to the rimonabant profile, CB_1_ antagonist activity of these ligands was highlighted through their counteraction to the increase of P-ERK 1/2 expression due to the effect of the reference cannabinoid agonist WIN 55,212-2. The results we have obtained with compounds **1b**–**d** are exemplified in Figure 2 (panels A and B) for derivative **1b**.

Different results have been instead detected, with derivative **1e** bearing a myrtanyl group in position 3 of the pyrazole ring. In this case, as for WIN 55,212-2, a dose response effect on P-ERK 1/2 expression was highlighted up to a concentration of **1e** of 75 nM (Figure 3). CB_1_ agonism of this new compound was confirmed through the detected profile from the competition study of P-ERK 1/2 expression in N1E-115 cells carried out with a pre-treatment step with the CB_1_ receptor antagonist rimonabant, followed by a 10 min exposure to derivative **1e** or the reference cannabinoid agonist WIN 55,212-2 (Figure 3).

#### 2.2.3. Isolated Organs (Mouse Vas Deferens)

The neutral antagonism of the new synthesized compounds **1b**–**d** was evaluated through an analysis of the profile of the electrically-evoked contractions on mouse vas deferens after a 30 min period of incubation with the different CB_1_ ligands. The analysis was carried out according to the previously reported procedure concerning other cannabinoid agents, rimonabant and corresponding analogues [21,22,23]. Compounds **1b**–**d** didn’t evoke significant changes compared to the basal contractile responses up to a concentration of 10^−6^ M, indicating a neutral antagonist profile. By contrast, the inverse agonist rimonabant elicited an increase of about 30% in the electrically evoked responses at a concentration of 10^−7^ M (data not shown).

Compound **1b** was further studied as a representative derivative of the subclass **1b**–**d**. No agonist activity was detected for **1b** following the amplitude of electrically evoked contractions of the vas deferens up to a concentration of 10^−6^ M. The CB_1_ antagonist behaviour of **1b** through the capability to antagonize the effect of the reference cannabinoid agonist WIN 55,212-2 has been highlighted in Figure 4. The inhibition of the contractions due to the activity of the reference cannabinoid agonist has been counteracted by **1b**, with a consequent shifting towards the right of the curves obtained with WIN 55,212-2 in the presence of the assayed compound if compared to that determined with WIN 55,212-2 alone. The antagonist potency of **1b**, expressed in terms of pA_2_, was 9.10. The obtained pA_2_ value is comparable to that of rimonabant (pA_2_ = 9.02; curve % of inhibition not shown) and of the lead compound **1a** (pA_2_ = 8.10; curve % of inhibition not shown), evidencing the potentiality of these 4,5-dihydrobenzo-oxa-cycloheptapyrazole derivatives for the treatments of diseases and disturbances related to CB_1_Rs.

According to the in vitro test results, an activation of CB_1_ receptors due to compound **1e** was highlighted through isolated organ assays. In particular, partial agonism, rather than a total agonism related profile, was detected for compound **1e**. In fact, this CB_1_ ligand produced concentration-related decreases in the amplitude of electrically-evoked contractions of the vas deferens, with a maximum inhibition around 60% rather than approximately 100% as for a total agonist as WIN 55,212-2 (Figure 5).

#### 2.2.4. In Vivo Assays: Gastrointestinal Transit (GIT)

Administration of the compound **1b** (0.5–1.0 mg/kg i.p.) produced an increase in gastrointestinal transit of up to approximately 65–70% in comparison to vehicle-treated mice (one-way ANOVA F(5,104) = 11.18, * *p* < 0.01). Results obtained with compound **1b** are reported in Figure 6a. Analogue results were obtained with derivatives **1c** and **1d** (data not shown). As shown in Figure 6b, the in vivo effect of the assayed compound was significantly blocked by the administration of cannabinoid reference agonist derivative WIN 55,212-2 (0.5 mg/kg i.p.). Considering the selectivity of the new compound for CB_1_R, the detected effect on GIT was attributed to this cannabinoid receptor subtype rather than to CB_2_.

#### 2.2.5. In Vivo Assays: Food Intake

It is well known that CB_1_Rs mediate the hyperphagic effect of cannabinoids. The selective CB_1_R inverse agonist rimonabant counteracts these effects, and, when administered alone, decreases standard chow intake and caloric consumption, presumably by antagonizing the actions of endogenously released endocannabinoids such as anandamide and 2-arachidonoylglycerol [24]. To test the potential anti-obesity effect of the new CB_1_ antagonist **1b** we performed food intake experiments according to the acute animal model described by Di Marzo [19]. For a comparison we also reported the results concerning the assumed lead compound of this new series, derivative **1a**.

The assayed compound significantly reduced food intake compared to the vehicle, starting from a dose of 3 mg/kg (Figure 7a). In spite of the difference of about one order of magnitude of the *K_i_*CB_1_ of both **1a** and **1b** compared to that of rimonabant, the induced decrease in food intake was analogous to that of the reference compound. Significant increments of the activity of **1a** and **1b** compared to that of the reference CB_1_ inverse agonist were highlighted at a dose of 10 mg/kg (Figure 7b), and this effect was sustained over time: indeed, both compounds still markedly decreased food intake at 2 h and 3 h, whereas the effect of rimonabant fades away. Moreover, no observable side effects were detected in the case of this new CB_1_ antagonist, even at 20 mg/kg. On the contrary, according to previously reported results by other authors [25,26], marked behavioural effects have been observed with rimonabant at the same dosage of 20 mg/kg, i.e., decreased frequency and duration of locomotion, rearing, grooming, sniffing and scratching.

Previously obtained results [9,10] and these latest results evidence the potentiality of these new derivatives in the control of appetite and in obesity treatment, with further potential application in the improvement of cardiovascular and metabolic risk factors. In fact, it has been demonstrated that the blocking of CB_1_Rs involves adiponectin up-regulation with improvement of cardiovascular risk factors and decrease of hyper-insulinemia and dyslipidemia, with reduction of obesity-associated hepatic steatosis [27,28]. Moreover, CB_1_ antagonists/inverse agonists significantly improve multiple cardiometabolic risk markers, inducing reductions in both intra-abdominal and liver fat in abdominally human obese patients [29].

### 2.3. Modelling

In order to try to relate the affinity and the activity data of the new compounds with their structural and geometrical features, a modelling study was performed in comparison with the reference compounds rimonabant and **1a** on their analogues **1b**–**e** and on a representative term of derivatives bearing an aryl substituent in position 3 of the pyrazole ring, compound **1f**. All the minimum energy conformations of the compounds were located through optimizations using DFT (density functional theory) methods at the B3LYP level [30,31] with the 6-311+G(d,p) basis set used for all atoms but chlorine, for which the 6-311+G(2df,p) basis set was used.

First, the results obtained for compound **1a** are described in detail. Its oxygenated heptacyclic ring can assume three geometries characterized by different folds of the ethyleneoxy chain and by particular values of the torsional angle that defines the deviation of the chlorophenyl and the pyrazole rings from planarity. In Figure 8 these three geometries (**1a-A**, **1a-B**, **1a-C**) are reported together with their relative energy values. The geometry of the substituent in position 3 of the pyrazole ring is characterized by a neat preference of the carboxamide function for the *s*-trans orientation with respect to pyrazole and by a piperidine ring in a chair conformation that can assume two different orientations, both of which are characterized by a mean plane of the heterocyclic ring almost perpendicular to the plane of the amide but differing for the orientation around the *N*-*N* bond that allows the ring nitrogen lone pair to point either in the opposite direction as the amide oxygen or in the same direction, the former geometry being preferred over the latter by 0.6–0.7 kcal/mol (see, for example, **1a-A** and **1a-D** in Figure 8).

The conformational behaviour of the substituent in position 3 of rimonabant is superimposable onto that of **1a** as they are characterized by the same piperazine ring. Their structural differences derive from the absence of the ethyleneoxy chain in rimonabant where the chlorophenyl group is connected by a single bond to pyrazole. Its minimum energy conformation is characterized by an angle of −53° between the two rings, even if it can deviate from this value at a very low energy cost. The crystal structure of the human CB_1_ cannabinoid receptor has been recently published [32,33]. The preferred conformation of compound **1a** (**1a-A**, Figure 8) is quite superimposable onto the geometry of rimonabant, as found in its favourite pose determined through docking calculations [32].

As expected, the tricyclic moiety of compounds **1b–f** shows exactly the same conformational behaviour as in **1a**. The structural differences in the substituent in position 3 of the pyrazole ring of compounds **1a**–**f** account for their geometrical and conformational differences. In **1b** the two located conformers resemble those of **1a**, except for the presence of a methine CH which replaces the ring nitrogen atom and for an opposite energy ranking (Figure 8 and Figure 9). In this case the geometry with the methine hydrogen atom pointing in the same direction as the carboxamide oxygen, **1b-D**, is preferred over the opposite one, **1b-A**, by about 2 kcal/mol.

The pyrrolidine of **1c** and the homopiperidine of **1d** behave quite similarly to the piperidine of **1a**. Two perpendicular orientations of the mean plane of the heterocyclic ring and that of the amide are found also in these cases (**1c-A** and **1c-D**, **1d-A** and **1d-D**, Figure 8) and the heterocyclic rings occupy about the same region of space (Figure 9).

Compound **1e**, bearing a *cis*-myrtanyl, shows a number of conformers, all of which present a large bicycloalkyl group, apart from the carboxamide plane. In some conformers this group is on one side but in other ones is on the other side of this plane (Figure 8), determining an unfavourable steric interaction with the residues surrounding the piperidine ring of rimonabant in the binding site of the CB_1_ receptor [32]. Hence, the *cis*-myrtanyl group in **1e** occupies a region of space different from that occupied by the (hetero)cycloalkyl moiety in **1a**–**d** (Figure 9), which accounts for the different agonist/antagonist behaviour of **1e** with respect to the other analogues.

Finally, the aryl compound **1f** shows its phenyl group lying on the same plane as the carboxamide function (Figure 8 and Figure 9), contrarily to the orthogonal arrangement of the (hetero)cycle found in **1a**–**d**, thus suggesting a possible explanation for the much lower affinity of the aryl analogues.

In Appendix A are reported the relative energy and selected geometrical data of conformers of compounds **1a**–**f**.

## 3. Experimental Section

### 3.1. General Procedures

Melting points were obtained on a Köfler melting point apparatus and are uncorrected. IR spectra were recorded as thin films (for oils) or nujol mulls (for solids) on NaCl plates with a Perkin Elmer 781 IR spectrophotometer and are expressed in ν (cm^−1^). All NMR spectra were taken on a Varian XL-200 NMR spectrometer with ^1^H and ^13^C being observed at 200 and 50 MHz respectively. Chemical shifts for ^1^H and ^13^C NMR spectra were reported in δ or ppm downfield from TMS [(CH_3_)_4_Si]. Multiplicities were recorded as s (singlet), br s (broad singlet), d (doublet), t (triplet), q (quartet), qu (quintuplet), dd (doublet of doudlets), and m (multiplet). Atmospheric pressure ionization electrospray (API-ES) mass spectra, when reported, were obtained using an Agilent 1100 series LC/MSD spectrometer. Combustion analyses on target compounds were performed and all compounds showed ≥95% purity. All reactions involving air or moisture-sensitive compounds were performed under an argon atmosphere. Flash chromatography was performed using Merck silica gel 60 (230–400 mesh ASTM). Thin layer chromatography (TLC) was performed with Polygram SIL N-HR/HV_254_ precoated plastic sheets (0.2 mm). The 3-chlorophenol **2** was purchased from Sigma-Aldrich.

#### 3.1.1. Synthesis of 4-(3-Chlorophenoxy)butyrric Acid (**3**)

To 3-chlorophenol (23.31 mmol, 1 equiv), NaOH pellets (1 equiv) were added and the whole was heated at 170 °C until all base was dissolved. To the mixture gamma-butyrrolactone (1.4 equiv) was added dropwise and the solution heated at the same temperature for 5 h, then poured into ice, acidified with 6N HCl and extracted with CHCl_3_, dried (using Na_2_SO_4_) and concentrated. The residue was purified by flash chromatography (petroleum ether/EtOAc 8:2) to give the acid **3** as a yellow solid. Yield: 2.34 g (46.87%). R_f_ = 0.15 (petroleum ether/EtOAc 8:2); m.p.: 47 °C (51–53 °C) [11]. IR (ν/cm^−1^): 3223 (OH), 1709 (CO). ^1^H-NMR (200 MHz, CDCl_3_): δ 2.11 (qu, 2H, *J* = 6.8 Hz), 2.58 (t, 2H, *J* = 7.4 Hz), 4.0 (t, 2H, J = 5.6 Hz), 6.76 (dd, 1H, J_m_ = 1.0 Hz, J_o_ = 8.2 Hz), 6.88 (s, 1H), 6.92 (d, 1H, J = 8.8 Hz), 7.18 (t, 1H, *J* = 7.8 Hz), 10.20 (br s, OH exch. with D_2_O).

#### 3.1.2. Synthesis of 8-Chloro-1-oxo-2,3,4,5-tetrahydrobenzocycloeptan-5-one (**4**)

A mixture of 4-(3-chlorophenoxy)butanoic acid **3** (27.96 mmol) and polyphosphoric acid (48 g) was heated at 90 °C for 1.5 h. The whole was poured into ice, extracted with CH_2_Cl_2_, washed (Na_2_CO_3_ aq. 10%), dried (using Na_2_SO_4_) and concentrated. The residue was purified by flash chromatography (petroleum ether/EtOAc 9:1) to give ketone **4** as an orange oil. Yield: 2.46 g (45.53%); b.p.: 46–47 °C/27 mm Hg (107 °C/0.4 Torr) [11]. ^1^H-NMR (200 MHz, CDCl_3_): δ 2.22 (qu, 2H, *J* = 6.4 Hz), 2.89 (t, 2H, *J* = 6.8 Hz), 4.25 (t, 2H, *J* = 6.6 Hz), 7.05–7.16 (m, 2H), 7.71 (d, 1H, *J* = 7.0 Hz).

#### 3.1.3. Synthesis of Ethyl 2-(8-Chloro-5-hydroxy-2,3-dihydrobenzo[*b*]oxepin-4-yl)-2-oxoacetate (**5**)

Sodium metal (2.0 equiv) was added in one small portion to dry EtOH (5 mL) and the mixture stirred until all the sodium had reacted. Ethyl oxalate (1 equiv) was added, followed by dropwise addition of a solution of ketone **4** (1 equiv, 6.0 mmol) in dry EtOH (30 mL). The solution was stirred at room temperature for 1.5 h and then was slowly poured into ice, after which 2N HCl was added. The solution was extracted with EtOAc, washed (H_2_O), dried (Na_2_SO_4_) and concentrated. The residue was purified by flash chromatography (petroleum ether/EtOAc 8:2) to give **5** as an orange solid. Yield: 1.58 g (88.74%). R_f_ = 0.46 (petroleum ether/ EtOAc 8:2); m.p.: 135 °C. IR (υ/cm^−1^): 1823 (CO), 1713 (CO), 1683 (CO). ^1^H-NMR (200 MHz, CDCl_3_/DMSO): δ 1.40 (t, 3H, *J* = 7.2 Hz), 2.78–2.98 (m, 2H), 4.22–4.50 (m, 4H), 7.08 (s, 1H), 7.15 (d, 1H, *J* = 8.2 Hz), 7.81 (d, 1H, *J* = 8.4 Hz), 15.76 (br s, OH exch. with D_2_O).

#### 3.1.4. Synthesis of Ethyl 8-chloro-1-(2,4-dichlorophenyl)-4,5-dihydrobenzo-1*H*-6-oxa-cyclohepta[1,2-*c*]pyrazole-3-carboxylate (**6**)

A stirred mixture of hydroxyketoester **5** (13.68 mmol, 1 equiv) and 2,4-dichlorophenylhydrazine hydrochloride (1.1 equiv) in EtOH (50 mL) was heated under reflux for 1.5 h. After concentration of the solvent, the analytically pure ester **6** was isolated by flash chromatography (petroleum ether/EtOAc 9:1) as an orange solid. Yield: 2.84 g (47.48%). R_f_ = 0.32 (petroleum ether/ EtOAc 9:1); m.p.: 135–136 °C. IR (υ/cm^−1^): 1712 (CO). ^1^H-NMR (200 MHz, CDCl_3_): δ 1.42 (t, 3H, *J* = 7.0 Hz), 3.44 (t, 2H, *J* = 5.4 Hz), 4.35–4.51 (m, 4H), 6.65 (d, 1H, *J* = 8.6 Hz), 6.81 (dd, 1H, *J*_m_ = 2.2 Hz, *J*_o_ = 8.6 Hz), 7.14 (s, 1H) 7.39–7.44 (m, 2H), 7.49 (s, 1H).

#### 3.1.5. Synthesis of 8-Chloro-1-(2,4-dichlorophenyl)-4,5-dihydrobenzo-1*H*-6-oxa-cyclohepta[1,2-*c*]pyrazole-3-carboxylic acid (**7**)

To a mixture of ester **6** (2 mmol, 1 equiv) in CH_3_OH (10 mL) was added a solution of potassium hydroxide in pellets (2 equiv) in CH_3_OH (7 mL) and a few drops of H_2_O. The resulting solution was heated under reflux for 12 h. The mixture was poured into ice and acidified with 1N HCl. The precipitate was filtered, washed with water and air-dried to yield the analytically pure acid **7** as a yellow solid. Yield: 0.75 g (91.23%). R_f_ = 0.38 (CHCl_3_/MeOH 9:1); m.p.: 230–231 °C; IR (υ/cm^−1^): 1689 (CO). ^1^H-NMR (200 MHz, CDCl_3_/DMSO): δ 3.10–3.45 (br s, 3H, 1 OH exch. with D_2_O), 4.30–4.48 (m, 2H), 6.67 (d, 1H, *J* = 8.4 Hz), 6.83 (dd, 1H, *J*_m_ = 3.0 Hz, *J*_o_ = 8.2 Hz), 7.13 (s, 1H), 7.44–7.50 (m, 3H).

#### 3.1.6. General Procedure for the Synthesis of Carbohydrazides (**1c**,**d**) and Carboxamides (**1e**–**j**)

A mixture of the 8-chloro-1-(2,4-dichlorophenyl)-4,5-dihydro-1*H*-6-oxa-cyclohepta[1,2-*c*]pyrazole-3-carboxylic acid **7** (6.4 mmol, 1 equiv), EDC (1.2 equiv) and HOBt (1.2 equiv) in dichloromethane (20 mL) was stirred at room temperature for 30 min. A solution of the requested amine or hydrazine (2 equiv) in dichloromethane (20 mL) was dropwise added and the whole was stirred at room temperature for 1–2 h. The solution was concentrated under reduced pressure and the analytically pure product was isolated after purification by flash chromatography and trituration with an appropriate solvent as indicated below.

#### 3.1.7. 8-Chloro-1-(2,4-dichlorophenyl)-*N*-pyrrolidin-1-yl-4,5-dihydrobenzo-1*H*-6-oxa-cyclohepta[1,2-*c*]pyrazole-3-carbohydrazide (**1c**) [20]

A general procedure was used to convert **7** and *N-*aminopyrrolidine hydrochloride into the title product. Because of an excess of the hydrochloride salt, an amount of triethylamine (2 equiv) was used in this reaction. The mixture was stirred for 2 h at room temperature, then purified by flash chromatography (petroleum ether/EtOAc 7:3) and triturated with petroleum ether to afford **1c** as a cream-coloured solid. Yield: 1.11 g (36.20%). R_f_ = 0.24 (petroleum ether/EtOAc 7:3); m.p.: 175 °C; IR (υ/cm^−1^): 3215 (NH), 1668 (CO). ^1^H-NMR (200 MHz, CDCl_3_): δ 1.80–1.95 (m, 4H), 2.95–3.05 (m, 4H), 3.45–3.55 (m, 2H), 4.30–4.40 (m, 2H), 6.60 (d, 1H, *J* = 9.2 Hz), 6.80 (d, 1H, *J* = 9.0 Hz), 7.15 (s, 1H), 7.24–7.40 (m, 3H), 7.53 (br s, 1H, NH exch. with D_2_O). ^13^C-NMR (50 MHz, CDCl_3_): δ 22.26 (CH_2_ × 2), 26.94 (CH_2_), 55.43 (CH_2_ × 2), 73.53 (CH_2_), 118.06 (C), 119.88 (C), 120.46 (C), 122.79 (CH), 123.45 (CH), 127.98 (CH), 128.35 (CH), 130.30 (CH), 130.69 (CH), 132.92 (C), 134.72 (C), 136.13 (C), 137.08 (C), 141.54 (C), 159.63 (C), 160.48 (CO). API-ES: calcd. for 477.77; found 476.00; anal. calcd, for C_22_H_19_Cl_3_N_4_O_2_: C, 55.31; H, 4.01; N, 11.73; found: C, 55.43; H, 4.02; N, 11.82.

#### 3.1.8. 8-Chloro-1-(2,4-dichlorophenyl)-*N*-homopiperidin-1-yl-4,5-dihydrobenzo-1*H*-6-oxa-cyclohepta[1,2-*c*]pyrazole-3-carbohydrazide (**1d**)

A general procedure was used to convert **7** and *N-*aminohomopiperidine into the title product. The mixture was stirred for 1.5 h at room temperature, then purified by flash chromatography (petroleum ether/EtOAc 7:3) to afford **1d** as a yellow solid. Yield: 1.14 g (35.29%). R_f_ = 0.31 (petroleum ether/EtOAc 7:3); m.p.: 137–138 °C. IR (υ/cm^−1^): 3289 (NH), 1678 (CO). ^1^H-NMR (200 MHz, CDCl_3_): δ 1.60–1.80 (m, 8H), 3.14 (t, 4H, *J* = 5.4 Hz), 3.49 (t, 2H, *J* = 5.2 Hz), 4.39 (t, 2H, *J* = 5.4 Hz), 6.60 (d, 1H, *J* = 8.8 Hz), 6.80 (dd, 1H, *J*_m_ = 2.2 Hz, *J*_o_ = 8.6 Hz), 7.15 (s, 1H), 7.30–7.38 (m, 3H), 7.54 (br s, 1H, NH, exch. with D_2_O). ^13^C-NMR (50 MHz, CDCl_3_): δ 26.22 (CH_2_ × 2), 26.92 (CH_2_ × 3), 58.37 (CH_2_ × 2), 73.56 (CH_2_), 116.05 (C), 119.93 (C), 120.46 (C), 122.78 (CH), 123.45 (CH), 127.99 (CH), 128.33 (CH), 130.30 (CH), 130.67 (CH), 132.88 (C), 134.69 (C), 136.07 (C), 137.07 (C), 138.99 (C), 143.74 (C), 159.95 (CO); API-ES: calcd. for 505.82; found 505.05; anal. calcd. for C_24_H_23_Cl_3_N_4_O_2_: C, 56.99; H, 4.58; N, 11.08; found: C, 57.13; H, 4.60; N, 11.13.

#### 3.1.9. 8-Chloro-1-(2,4-dichlorophenyl)-*N*-(6,6-dimethyl-bicyclo[3.1.1]hept-2-yl-methyl)-4,5-dihydrobenzo-1*H*-6-oxa-cyclohepta[1,2-c]pyrazole-3-carboxamide (**1e**)

A general procedure was used to convert **7** and *cis-*myrtanylamine into the title product. The mixture was stirred for 2 h at room temperature, then purified by flash chromatography (petroleum ether/EtOAc 9:1) to afford **1e** as a white solid. Yield: 0.89 g (25.47%). R_f_ = 0.31 (petroleum ether/EtOAc 9:1); m.p.: 142–143 °C; IR (υ/cm^−1^): 3403 (NH), 1676 (CO). ^1^H-NMR (200 MHz, CDCl_3_): δ 1.07 (s, 3H), 1.19 (s, 3H), 1.45–1.62 (m, 1H), 1.82–2.04 (m, 6H), 2.21–2.43 (m, 2H), 3.26–3.57 (m, 4H), 4.35–4.42 (m, 2H), 6.60 (d, 1H, *J* = 8.6 Hz), 6.81 (d, 1H, *J* = 8.6 Hz), 6.93 (t, 1H, NH, exch. with D_2_O), 7.14 (s, 1H), 7.31–7.39 (m, 2H), 7.54 (s, 1H). ^13^C-NMR (50 MHz, CDCl_3_) δ 19.84 (CH_3_), 23.22 (CH_3_), 26.00 (CH_2_), 27.08 (CH_2_), 27.96 (CH_3_), 33.26 (CH_2_), 38.70 (C), 41.32 (CH), 41.50 (CH), 43.80 (CH), 44.57 (CH_2_), 73.56 (CH_2_), 120.03 (C × 2), 122.75 (CH), 123.44 (CH), 128.01 (CH), 128.33 (CH), 130.32 (CH), 130.67 (CH), 132.93 (C), 134.61 (C), 136.05 (C), 137.16 (C), 139.05 (C), 144.51 (C), 159.62 (C), 162.25 (CO); API-ES: calcd. for 544.90; found 544.20; anal. calcd. for C_28_H_28_Cl_3_N_3_O_2_: C, 61.72; H, 5.18; N, 7.71; found: C, 61.86; H, 5.21; N, 7.73.

#### 3.1.10. 8-Chloro-1-(2,4-dichlorophenyl)-*N*-phenyl-4,5-dihydrobenzo-1*H*-6-oxa-cyclohepta[1,2-*c*]pyrazole-3-carboxamide (**1f**)

A general procedure was used to convert **7** and alanine into the title product. The mixture was stirred for 1 h at room temperature, then purified by flash chromatography (petroleum ether/EtOAc 9:1) to afford **1f** as a yellowish solid. Yield: 1.35 g (43.48%). R_f_ = 0.28 (petroleum ether/EtOAc 9:1); m.p.: 130 °C. IR (υ/cm^−1^): 3386 (NH), 1688 (CO). ^1^H-NMR (200 MHz, CDCl_3_): δ 3.57 (t, 2H, *J* = 6.0 Hz), 4.42 (t, 2H, *J* = 6.0 Hz), 6.69 (d, 2H, *J* = 8.4 Hz), 6.82 (d, 1H, *J* = 8.4 Hz), 7.12–7.20 (m, 3H), 7.32–7.42 (m, 3H), 7.67 (d, 2H, *J* = 7.6 Hz), 8.76 (br s, 1H, NH, exch. with D_2_O). ^13^C-NMR (50 MHz, CDCl_3_): δ 27.22 (CH_2_), 73.41 (CH_2_), 119.72 (CH × 2), 119.79 (C), 120.48 (C), 122.80 (CH), 123.54 (CH), 124.16 (CH), 128.05 (CH), 128.40 (CH), 129.00 (CH × 2), 130.29 (CH), 130.75 (CH), 132.97 (C), 134.85 (C), 136.29 (C), 137.06 (C), 137.66 (C), 139.49 (C), 144.31 (C), 159.78 (C), 160.13 (CO); API-ES: calcd. for 484.76; found 484.10. Anal. calcd. for C_24_H_16_Cl_3_N_3_O_2_: C, 59.46; H, 3.33; N, 8.67; found: C, 59.65; H, 3.34; N, 8.70.

#### 3.1.11. 8-Chloro-1-(2,4-dichlorophenyl)-*N*-(p-chlorophenyl)-4,5-dihydrobenzo-1*H*-6-oxa-cyclohepta[1,2-*c*]pyrazole-3-carboxamide (**1g**)

A general procedure was used to convert **7** and *p*-chloroaniline into the title product. The mixture was stirred for 1 h at room temperature, then purified by flash chromatography (petroleum ether/EtOAc 8:2) to afford **1g** as a green solid. Yield: 0.58 g (17.39%). R_f_ = 0.37 (petroleum ether/EtOAc 8:2); m.p.: 239 °C. IR (υ/cm^−1^): 3388 (NH), 1688 (CO). ^1^H-NMR (200 MHz, CDCl_3_): δ 3.55 (t, 2H, *J* = 6.4 Hz), 4.42 (t, 2H, *J* = 6.0 Hz), 6.64 (d, 1H, *J* = 8.4 Hz), 6.82 (dd, 1H, *J*_m_ = 2.2 Hz, *J*_o_ = 8.4 Hz), 7.16 (s, 1H), 7.31 (d, 2H, *J* = 8.8 Hz), 7.37–7.41 (m, 3H), 7.63 (d, 2H, *J* = 9.0 Hz), 8.76 (br s, 1H, NH, exch. with D_2_O). ^13^C-NMR (50 MHz, CDCl_3_): δ27.20 (CH_2_), 73.34 (CH_2_), 119.70 (C), 120.51 (C), 120.88 (CH × 2), 122.83 (CH), 123.58 (CH), 128.05 (CH), 128.43 (CH), 129.02 (CH × 2), 130.26 (CH), 130.77 (CH), 132.97 (C), 134.95 (C), 136.28 (C), 136.38 (C), 137.00 (C), 139.61 (C), 141.54 (C), 144.08 (C), 159.81 (C), 160.09 (CO). API-ES: calcd. for 519.21; found 518.80. Anal. calcd. for C_24_H_15_Cl_4_N_3_O_2_: C, 55.52; H, 2.91; N, 8.09; found: C, 55.60; H, 2.92; N, 8.11.

#### 3.1.12. 8-Chloro-1-(2,4-dichlorophenyl)-*N*-tolyl-4,5-dihydrobenzo-1*H*-6-oxa-cyclohepta[1,2-*c*] pyrazole-3-carboxamide (**1h**)

A general procedure was used to convert **7** and *p*-toluidine into the title product. The mixture was stirred for 1 h at room temperature, then purified by flash chromatography (petroleum ether/ EtOAc 8:2) to afford **1h** as a yellow solid. Yield: 0.89 g (27.90%). R_f_ = 0.33 (petroleum ether/EtOAc 8:2); m.p.: 181 °C. IR (υ/cm^−1^): 3383 (NH), 1688 (CO). ^1^H-NMR (200 MHz, CDCl_3_): δ 2.33 (s, 3H), 3.56 (t, 2H, *J* = 6.4 Hz), 4.41 (t, 2H, *J* = 6.4 Hz), 6.64 (d, 1H, *J* = 8.8 Hz), 6.82 (d, 1H, *J* = 8.6 Hz), 7.13–7.18 (m, 3H), 7.34–7.44 (m, 2H), 7.53–7.57 (m, 3H), 8.69 (br s, 1H, NH, exch. with D_2_O). ^13^C-NMR (50 MHz, CDCl_3_) δ 20.86 (CH_3_), 27.22 (CH_2_), 73.41 (CH_2_), 119.70 (CH), 119.72 (C), 119.79 (C), 120.48 (C), 122.80 (CH), 123.54 (CH), 124.16 (CH), 128.05 (CH), 128.40 (CH), 129.00 (CH × 2), 130.29 (CH), 130.75 (CH), 132.97 (C), 134.85 (C), 136.29 (C), 137.06 (C), 137.66 (C), 139.49 (C), 144.31 (C), 159.78 (C), 160.13 (CO). API-ES: calcd. for 498.79; found 498.20. Anal. calcd. for C_25_H_18_Cl_3_N_3_O_2_: C, 60.20; H, 3.64; N, 8.42; found: C, 60.29; H, 3.65; N, 8.45.

#### 3.1.13. 8-Chloro-1-(2,4-dichlorophenyl)-*N*-(p-methoxyphenyl)-4,5-dihydrobenzo-1*H*-6-oxa-cyclohepta[1,2-*c*]pyrazole-3-carboxamide (**1i**)

A general procedure was used to convert **7** and *p*-methoxyaniline into the title product. The mixture was stirred for 1 h at room temperature, then purified by flash chromatography (petroleum ether/ EtOAc 7:3) to afford **1i** as an orange solid. Yield: 0.57 g (17.40%). R_f_ = 0.27 (petroleum ether/EtOAc 8:2); m.p.: 214 °C. IR (υ/cm^−1^): 3378 (NH), 1682 (CO). ^1^H-NMR (200 MHz, CDCl_3_): δ 3.53–3.59 (m, 2H), 3.80 (s, 3H), 4.40–4.44 (m, 2H), 6.64 (d, 1H, *J* = 8.6 Hz), 6.79–6.91 (m, 3H), 7.16 (s, 1H), 7.38–7.41 (m, 2H), 7.56–7.60 (m, 3H), 8.65 (br s, 1H, NH, exch. with D_2_O). ^13^C-NMR (50 MHz, CDCl_3_) δ 27.19 (CH_2_), 55.47 (CH_3_), 73.43 (CH_2_), 114.14 (CH × 2), 119.83 (C), 120.39 (C), 121.46 (CH × 2), 122.79 (CH), 123.27 (C), 123.51 (CH), 128.05 (CH), 128.40 (CH), 130.29 (CH), 130.73 (CH), 130.78 (C), 132.97 (C), 134.81 (C), 136.25 (C), 137.08 (C), 144.41 (C), 156.29 (C), 159.76 (C), 159.97 (CO). API-ES: calcd. for 514.79; found 514.10. Anal. calcd. for C_25_H_18_Cl_3_N_3_O_3_: C, 58.33; H, 3.52; N, 8.16; found: C, 58.42; H, 3.53; N, 8.19.

#### 3.1.14. 8-Chloro-1-(2,4-dichlorophenyl)-*N*-(p-trifluoromethylphenyl)-4,5-dihydrobenzo-1*H*-6-oxa-cyclohepta[1,2-*c*]pyrazole-3-carbohydrazide (**1j**)

A general procedure was used to convert **7** and *p*-trifluoroaniline into the title product. The mixture was stirred for 1 h at room temperature, then purified by flash chromatography (petroleum ether/EtOAc 9:1) to afford **1j** as a yellow solid. Yield: 1.39 g (39.36%). R_f_ = 0.20 (petroleum ether/EtOAc 9:1); m.p.: 249–250 °C. IR (υ/cm^−1^): 3379 (NH), 1698 (CO). ^1^H-NMR (200 MHz, CDCl_3_): δ 3.53–3.57 (m, 2H), 4.40–4.44 (m, 2H), 6.65 (d, 1H, *J* = 8.4 Hz), 6.84 (d, 1H, *J* = 8.6 Hz), 7.17 (s, 1H), 7.34–7.46 (m, 2H), 7.57–7.63 (m, 3H), 7.80 (d, 2H, *J* = 8.4 Hz), 8.92 (br s, 1H, NH, exch. with D_2_O). ^13^C-NMR (50 MHz, CDCl_3_) δ 27.22 (CH_2_), 73.29 (CH_2_), 119.23 (CH × 2), 119.63 (C), 120.62 (C), 122.85 (CH), 123.61 (CH), 124.84 (C), 126.25 (CH), 126.32 (CH), 128.04 (CH), 128.46 (CH), 130.24 (CH), 130.78 (CH), 132.97 (C), 135.03 (C), 136.46 (C), 136.96 (C), 139.73 (C), 140.76 (C), 141.55 (C), 143.89 (C), 159.84 (C), 160.28 (CO). API-ES: calcd. for 552.76; found 552.20. Anal. calcd. for C_25_H_15_Cl_3_F_3_N_3_O_2_: C, 54.32; H, 2.75; N, 7.60; found: C, 54.41; H, 2.74; N, 7.63.

### 3.2. Evaluation of Biological Activity

#### 3.2.1. Chemicals and Drugs for in Vitro and in Vivo Assays

[^3^H]-CP-55,940 (specific activity 180 Ci/mmol) was purchased from Perkin Elmer Italia. CP-55,940 and WIN 55,212-2 were obtained from Tocris Cookson Ltd. (Bristol, UK). *N-*piperidinyl-5-(4-chlorophenyl)-1-(2,4-dichlorophenyl)-4-methyl-1*H*-pyrazole-3-carboxamide (rimonabant) was purchased from KEMPROTECH Limited, Middlesbrough, UK.

For both radioreceptor binding and isolated organ experiments, drugs were dissolved in dimethyl-sulfoxide (DMSO). DMSO concentration in the different assays never exceeded 0.1% (*v*/*v*) and was without effects.

In vivo assays were carried out by solubilizing compounds in saline solution containing Tween 80 and Ethanol (both at 2.5 wt%). Microemulsion systems of the compounds have been obtained. All animal experiments were performed according to the EU guidelines for the care and use of experimental animals (CEE N° 86/609). Animals (Charles River, Calco, LC, Italy) were housed in animal care quarters; temperatures were maintained at 22 ± 2 °C (humidity 55 ± 5%) on a 12 h light/dark cycle. Food and water were available ad libitum before starting with the experimental procedures.

#### 3.2.2. Radioligand Binding Methods

Male CD1 mice weighing 30–35 g were killed by cervical dislocation and the brain (minus cerebellum) and spleen were rapidly removed and frozen. After thawing, tissues were homogenated in 20 vol. (*wt/v*) of ice-cold TME buffer (50 mM Tris-HCl, 1 mM EDTA and 3.0 mM MgCl_2_, pH 7.4). The homogenates were centrifuged at 1,086 ×*g* for 10 min at 4 °C, and the resulting supernatants were centrifuged at 45,000 ×*g* for 30 min.

[^3^H]-CP-55,940 binding was performed using the method previously described by Ruiu et al. [16]. Briefly, the membranes (30–80 µg of protein) were incubated with 0.5–1 nM of [^3^H]-CP-55,940 for 1 h at 30 °C in a final volume of 0.5 mL of TME buffer containing 5 mg/mL of fatty acid-free bovine serum albumin. Non-specific binding was estimated in the presence of 1 µM of CP-55,940. All binding studies were performed in disposable glass tubes pre-treated with Sigma-Cote (Sigma Chemical Co., Ltd., Poole, UK), in order to reduce non-specific binding. The reaction was terminated by rapid filtration through Whatman GF/C filters presoaked in 0.5% polyethyleneimine (PEI) using a Brandell 36-sample harvester (Gaithersburg, MD, USA). Filters were washed five times with 4 mL aliquots of ice-cold Tris HCl buffer (pH 7.4) containing 1 mg/mL BSA. The filter bound radioactivity was measured in a liquid scintillation counter (Tricarb 2900, Packard, Meridien, ID, USA) with 4 mL of scintillation fluid (Ultima Gold MV, Packard).

Protein determination was performed by means of a Bradford protein assay using BSA as a standard according to the protocol of the supplier (Bio-Rad, Milan, Italy) [34].

All experiments were performed in triplicate and results were confirmed in at least five independent experiments. Data from radioligand inhibition experiments were analysed by nonlinear regression analysis of a Sigmoid Curve using Graph Pad Prism program. IC_50_ values were derived from the calculated curves and converted to *K*_i_ values as previously described [35].

#### 3.2.3. In Vitro Assays

N1E-115 cells were grown at 37 °C in humidified 5% CO_2_ in Dulbecco’s Modified Eagle’s Medium (high glucose supplemented with 10% fetal bovine serum (FBS), 1 μg/mL penicillin-streptomycin, 2 mM l-glutamine, 2.5 μg/mL amphotericin B and 50 μg/mL gentamicin). N1E-115 cells were treated with the compounds to be assayed and reference compounds WIN55,212-2 (cannabinoid agonist) and rimonabant (CB_1_ selective antagonist). Tested and reference compounds were dissolved in Dulbecco’s phosphate buffered saline (D-PBS) with 1% DMSO and treatments were performed in a volume of 10 μL/mL of cell suspension.

Cells lines from the European Collection of Cell Cultures (ECACC), growth medium, FBS, penicillin-streptomycin, l-glutamine, amphotericin B, gentamicin, D-PBS and DMSO were purchased from Sigma-Aldrich (Milano, Italy).

P-ERK 1/2 expression was determined through Western blot analysis according to the previously reported procedure [36]. One-way ANOVA was performed as a statistical analysis using the Graph Pad Prism program (San Diego, CA, USA).

To verify whether the observed effects were CB_1_ specific, a 5 min pre-treatment with the CB_1_ receptor antagonist rimonabant (1 μM) was carried out before exposure to the compounds that displayed a significant induction of P-ERK (as shown by Western blot analysis). The concentration of rimonabant was chosen among the concentrations lacking an intrinsic activity, providing an appropriate block of the CB_1_ receptor. The same procedure was adopted to determine CB_1_ specific antagonism for compounds which had no effect on P-ERK 1/2 expression themselves. In these last cases, WIN55,212-2 at 25 nM as a reference cannabinoid agonist agent and the new CB_1_ antagonists at 150–200 nM were employed.

#### 3.2.4. Isolated Organs

Vas deferens was obtained from male albino CD1 mice weighing 30 to 40 g. Tissue was mounted in a 10 mL organ bath at an initial tension of 0.5 g using a previously described method [16]. The bath contained Krebs-Henseleit solution (118.2 mM NaCl, 4.75 mM KCl, 1.19 mM KH_2_PO_4_, 25.0 mM NaHCO_3_, 11.0 mM glucose, and 2.54 mM CaCl_2_), which was kept at 37 °C and bubbled with 95% O_2_ and 5% CO_2_. Isometric contractions were evoked by stimulation with 0.5 sec trains of three pulses of 110% maximal voltage (train frequency, 0.1 Hz; pulse duration, 0.5 msec) through platinum electrodes attached to the upper end of each bath and a stainless-steel electrode attached to the lower end. Stimuli were generated using a Grass S88K stimulator and then amplified (multiplexing pulse booster 316S; Ugo Basile, Comerio, Italy) and divided to yield separate outputs to four organ baths. Contractions were monitored using a computer equipped with a data recording and analysis system (PowerLab 400) linked via preamplifiers (QuadBridge) to an F10 transducer (Biological Instruments, Besozzo, Italy). Each tissue was subject to several periods of stimulation. The first of these stimulation periods began after the tissue had equilibrated in the buffering medium but before drug administration, and continued for 10 min. The stimulator was then switched off for 15 min, after which the tissues were subjected to further periods of stimulation each lasting 5 min and separated by a stimulation-free period. The compounds to be assayed were added once the contractile responses to electrical stimulation were reproducible. Preparations were exposed to cumulative increasing concentrations of the compounds to obtain concentration-response curves in the case of CB_1_ agonist (and partial agonist) compounds. When no inhibition action on contractile response was evidenced, CB_1_ antagonism was evaluated by carrying out experiments with the reference cannabinoid agonist WIN 55,212-2. The compounds to be assayed were added at a fixed concentration 20 min before the first concentration of WIN 55,212-2. Inhibition of the electrically evoked twitch response was expressed in percentage terms and was calculated by comparing the amplitude of the twitch response after each addition of the agonist compound (alone or in the presence of a potential CB_1_ antagonist) with the amplitude detected in absence of the compounds to be assayed (basal value). The pA_2_ values for competitive antagonism were calculated by Schild regression analysis [37]. Data were plotted as log antagonist concentration versus log [(antagonist concentration/agonist concentration)-1].

#### 3.2.5. Gastrointestinal Transit

The same procedure previously reported by Murineddu et al. [17] was adopted to evaluate the effect of the compounds on GIT. CD1 mice weighing 30–35 g were used. Carmine (Sigma Chemical Co, St. Louis, MO, USA) was suspended (6% *wt*/*v*) in distilled water containing 0.5% methylcellulose and administered by gavage at the dose of 0.3 mL/mouse. Briefly, after 20 min from carmine administration, mice were sacrificed by cervical dislocation and the stomach and small intestine were removed and the omentum was separated, avoiding stretching. The length of the intestine was measured starting from the pyloric sphincter to the ileocecal junction. The distance travelled by the head of the marker was measured and expressed as a percent of the total length of the small intestine. Using this procedure, the marker was consistently found at approximately 50% of the small intestine, 20 min after its administration to naive mice.

WIN 55,212-2 and compounds to be assayed were administered i.p. 20 min before the i.g. administration of the marker. 20 min later, mice were killed by dislocation cervical and intestines were removed. In the antagonism test, the assayed compounds were administered i.p. 30 min prior to the i.g. administration of the marker and 10 min prior to the i.p. injection of WIN 55,212-2 (0.5 mg/kg). The length of the intestine from the pyloric sphincter to the ileocecal junction and the distance travelled by the carmine front were both measured and recorded. Groups of n = 10 mice were used in each experiment.

Statistical analyses were performed by one-way ANOVA followed by Newman-Keuls post-hoc comparison.

#### 3.2.6. Food Intake

A widely described method was used for the screening of the new CB_1_ antagonists in the food intake animal model [19]. C57Bl6/J mice weighing 20–25 g were housed individually on a reverse light-dark cycle in a room with temperature and humidity control. The mice were fasted for 18 h before testing. Vehicle or CB_1_ antagonists were administered by an intraperitoneal injection and 15 min later a known amount of food was presented. Food pellets were weighed after each hour for three hours. Rimonabant was used as internal positive control in order to evaluate the efficacy of each compound. All animal experiments were performed according to the UE (CEE N° 86/609) guidelines for the care and use of experimental animals.

Two-way ANOVA with Bonferroni post-testing was performed using GraphPad Prism version 5.00 for Windows (GraphPad Software, San Diego CA, USA).

### 3.3. Computational Methods

All calculations were carried out using the Gaussian 09 [38] program package. The conformational space of the compounds was explored through optimizations at the B3LYP level [30,31] with the 6-311+G(d,p) basis set (6-311+G(2df,p) for the chlorine atoms). First, a starting geometry was optimized. Then, the energy profiles for rotation around the single bonds of the substituent in position 3 of pyrazole, as well as of the aryl groups, were determined with a step size of 30°, and ring flexibility of the tricyclic moiety was considered. All the combinations of the observed minima were used to generate starting geometries optimized as above. Vibrational frequencies were computed at the same level of theory to verify that the optimized structures were minima.

## 4. Conclusions

Ten novel 4,5-dihydrobenzo-oxa-cycloheptapyrazole carboxamides were synthesized. They were all found to bind to cannabinoid receptors and those bearing a (hetero)cyclic amine and a myrtanyl group in the C_3_-carboxamide moiety bind with high affinities to CB_1_ receptors as compared to aryl amides analogues. In particular, compound **1a** revealed the highest CB_1_ affinity, whereas derivative **1b** showed the highest CB_1_ selectivity, demonstrating that aliphatic carboxamides seem to be the optimum for CB_1_ receptors.

Interestingly, whereas carbamoyl derivatives with an aliphatic monocyclic ring **1b**–**d** act as antagonists at the CB_1_ receptor according to parent compound **1a**, as evidenced by both in vitro and in ex-vivo assays and further confirmed by in vivo tests, the introduction in the C_3_-carboxamide position of a bicyclic bulky group as myrtanyl was found to be responsible for the partial agonist activity of **1e**, as confirmed by both in vitro studies and isolated organ assays. Modelling studies evidenced that the large bicycloalkyl group of **1e** occupies a region of space different from that occupied by the (hetero)cycloalkyl moiety in **1a**–**d** and this could account for the different agonist/antagonist behaviour of **1e** with respect to the other analogues.

CB_1_ affinity of the new derivatives was also lower than that of the reference CB_1_ inverse agonist rimonabant. However, despite the differences in K*_i_*CB_1_ values, it is important to note that the effect on food intake of the compound with the highest CB_1_ affinity among the studied class, compound **1b**, was comparable or significantly more marked than that obtained with the reference compounds rimonabant and **1a**, respectively. The potentiality of the compound **1b** has been further supported by the absence of observable side effects, even at the highest assayed dosage.

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
