# Peer review of "Development of Oxygen-Bridged Pyrazole-Based Structures as Cannabinoid Receptor 1 Ligands"

_molecules, 2019, doi:10.3390/molecules24091656_

Round 1
Reviewer 1 Report
Murineddu et al. have synthesized a new series of cannabinoid ligands, mainly CB1R antagonists, based in a previous molecule. The biological activity of the new compounds has been investigated in relevant in vitro, isolated organ assays and modeling, and they describe the structure activity studies. Of note, the selectivity and potency of some of the compounds, that behaved as CB1R neutral antagonists. They also studied the effects of the lead compound on food intake and gastrointestinal transit. The results suggest their potential use in the treatment of obesity.
The work has been well performed, the assays are very appropriate, and several doses were studied in experiments in vivo. The manuscript (MS) is well written and easy to read, and the recent literature is used properly. The results are interesting and they have biomedical interest.
Minor comments:
-Line 80. “individuate” it is not clear its meaning: would be “disclose” or “reveal”? Please change.
-Double check the general structure at the beginning of Table 1.
-Fig 2, page 7, A and B are missing in the panels. Revise the text accordingly (lines 175-179). Change dose by concentration in the legend to Fig. 2, and in lines 581 and 582 as well. Double check titles of treatment groups in Fig. 2 B.
-Line 280. “Similar” is more appropriate than “analogue” in that context.
-In regard to the effect of 1b and 1a at the 10 mg/kg dose on food intake it should be mentioned that it is sustained over time when compared with rimonabant. Indeed, both compounds still markedly decreased food intake at 2h and 3h while the effect of rimonabant fades away.
-Line 544. It looks like a sentence is missing, since after placing the tissues on a cold plate in the next sentence the samples are thawed. Likely the samples were frozen until assayed. Please double check.
-The sentence spanning lines 671-672 is difficult to read and should be rewritten.
“…it is important to note that the effect on food intake of the highest CB1 affine compound among the studied class…”. Did the authors meant “…it is important to note that the effect on food intake of the compound with the highest CB1 affinity among the studied class…”.
Author Response
Minor comments:
1) Line 80. “individuate” it is not clear its meaning: would be “disclose” or “reveal”? Please change.
Authors’ answer: Done (disclose).
2) Double check the general structure at the beginning of Table 1.
Authors’ answer: The Q substituent in general structure has been replaced by R, in accordance with the column B of the table 1.
3) Fig 2, page 7, A and B are missing in the panels. Revise the text accordingly (lines 175-179).
Authors’ answer: Done.
Change dose by concentration in the legend to Fig. 2, and in lines 581 and 582 as well.
Authors’ answer: Done, also in the legend of figure 3, and in lines 601 and 602 (not 581-582).
Double check titles of treatment groups in Fig. 2 B.
Authors’ answer: The figure has been corrected.
4) Line 280. “Similar” is more appropriate than “analogue” in that context.
Authors’ answer: Done in line 261 (not 280)
5) In regard to the effect of 1b and 1a at the 10 mg/kg dose on food intake it should be mentioned that it is sustained over time when compared with rimonabant. Indeed, both compounds still markedly decreased food intake at 2h and 3h while the effect of rimonabant fades away.
Authors’ answer: Done: the sentence “…(Figure 7B) and this effect is sustained over the time: indeed, both compounds still markedly decreased food intake at 2h and 3h, whereas the effect of rimonabant fades away. Moreover…” has been added in lines 295-296.
6) Line 544. It looks like a sentence is missing, since after placing the tissues on a cold plate in the next sentence the samples are thawed. Likely the samples were frozen until assayed. Please double check.
Authors’ answer: The sentence “Male CD1 mice weighing 30-35 g were killed by cervical dislocation and the brain (minus cerebellum) and spleen were rapidly removed and placed on an ice cold plate.” has been corrected as follows: ” … were rapidly removed and frozen.”
7) The sentence spanning lines 671-672 is difficult to read and should be rewritten.
Authors’ answer: The sentence “Then, the energy profiles for rotation around the single bonds of the substituent in position 3 of pyrazole as well as of the aryl groups were determined…“ has been corrected as follow: “…position 3 of pyrazole, as well as of the aryl groups, were determined… “. (now in lines 684-686)
8) “…it is important to note that the effect on food intake of the highest CB1 affine compound among the studied class…”. Did the authors meant “…it is important to note that the effect on food intake of the compound with the highest CB1 affinity among the studied class…”.
Authors’ answer: The sentence: “…effect on food intake of the highest CB1 affine compound among…” has been corrected as suggested: “…on food intake of the compound with the highest CB1 affinity among…” (line 707).

Reviewer 2 Report
The article by Murineddu et al is proposing the pharmacomodulation of CB1R antagonist firstly reported by the author in 2013 and 2016 and patented in 2010. On the basis for the first structure of compound 1a and 1b which could be considered as rigidified analogs of rimonabant, novel pharmacomodulation are proposed.
The main objective of these modulations is to obtain compound unable to cross the BBB in order to limit adverse effects. The authors are proposing the introduction of other cycloalky or aromatic moieties.
If the objective is to reduce the lipophilicity to limit the ability of the compound to cross the BBB, the introduction of an aromatic moiety might not be ideal. Calculated LogP could be interesting in this aspect. In the same idea and in comparison with 1a, introduction of phenyl hydrazine would be interesting.
In order to justify the impact of these modulation on activity and selectivity the authors could also use the crystal structure of CB1 and CB2 receptors published recently. Some of the following papers are describing the interaction of Rimonabant within the active site which could be valuable information for this study.
Crystal structure of Rimonabant in CB1 binding pocket: Hua T. et al Cell. 2016 167(3): 750–762.e14.
Docking from crystal structure Z. Shao, et al. Nature 2016, 540, 602–606
Considering that none of the novel compounds show significant improvement and that most of the in vivo evaluation of already published 1a and 1B, I recommend the justification of the modulation to be clearly justified in order to be published in Molecules.
A list of modifications could be find below.
Pyrrolidine 1c has already been described in the following patent By Lohray, Braj Bhushan; Lohray, Vidya Bhushan; Srivastava, Brijesh From U.S. (2011), US 8030345 B2 20111004. The patent should be cited.
Minor corrections:
- P1 l42: Replace Rimonbant by Rimonabant
- P2 Scheme 1: include the structure of rimonabantStructure of Imiquimod and resiquimod should be added in Figure 1.
- P3 line 90 and 94: Replace –butyrrolactone by gamma-butyrolactone
- P3 line 92: Replace BOHt by HOBt
- P3 line 92: Replace Q-NH2 by R-NH2
- P3 line 101: Replace amine by amine or hydrazine.
- P4 line 141: Replace Q by R in the general formula of Table 1
- P5 Align Cmpd number with Ki Values (From 1d to 1j)
- P11 line 260 What is the concentration of 1b in the third colum (0.05?)
- Numbering of compound is a bit confusing between compound 6 and 14, 7 and 15, 8 and 16, 9 and 17, and 10 and 18 respectively. This should be simplified based of the similarity of the compound. 6 as 6a and 14a-c as 6b-d for example.
- Cytotoxicity exploration using MTT assay should be added in SI.
Author Response
The article by Murineddu et al is proposing the pharmacomodulation of CB1R antagonist firstly reported by the author in 2013 and 2016 and patented in 2010. On the basis for the first structure of compound 1a and 1b which could be considered as rigidified analogs of rimonabant, novel pharmacomodulation are proposed.
The main objective of these modulations is to obtain compound unable to cross the BBB in order to limit adverse effects. The authors are proposing the introduction of other cycloalky or aromatic moieties.
If the objective is to reduce the lipophilicity to limit the ability of the compound to cross the BBB, the introduction of an aromatic moiety might not be ideal. Calculated LogP could be interesting in this aspect. In the same idea and in comparison with 1a, introduction of phenyl hydrazine would be interesting.
Authors’ answer: Thanks for your suggestion. In previous papers we reported that results obtained from radioligand binding experiments on similar compounds suggest that substitution of the cycloalkyl carbamoyl groups with a phenyl carbamoyl moiety led to decrease of affinity. Moreover, we founded that substitution of the carbamoyl moiety with the phenylhydrazide unit was detrimental for the CBR affinity. In the light of these previously reported considerations, and due to the binding values for novel compounds bearing an aromatic moiety on the amide function, in this work we did not consider the introduction of the arylhydrazine group.
In order to justify the impact of these modulation on activity and selectivity the authors could also use the crystal structure of CB1 and CB2 receptors published recently. Some of the following papers are describing the interaction of Rimonabant within the active site which could be valuable information for this study.
Crystal structure of Rimonabant in CB1 binding pocket: Hua T. et al Cell. 2016 167(3): 750–762.e14.
Docking from crystal structure Z. Shao, et al. Nature 2016, 540, 602–606
Authors’ answer: The following sentences and references have been added: “The crystal structure the human CB1 cannabinoid receptor was recently published [32,33]. The preferred conformation of compound 1a (1a-A, Figure 8) is quite superimposable to the geometry of rimonabant as found in its favourite pose determined through docking calculations [32].” in lines 343-345, and “…(Figure 8), determining an unfavourable steric interaction with the residues surrounding the piperidine ring of ribonabant in the binding site of the CB1 receptor [32]. So, the cis-myrtanyl group in 1e occupies a region…” in lines 363-365.
Considering that none of the novel compounds show significant improvement and that most of the in vivo evaluation of already published 1a and 1B, I recommend the justification of the modulation to be clearly justified in order to be published in Molecules.
Authors’ answer: By introducing different substituents in the tricyclic oxygenated structure we wanted to confirm that the presence of aliphatic substituents on carboxamide function is pivotal for the CB1R affinity. The introduction of three aliphatic susbtituents as pyrrolidine (1c), homopiperidine (1d) and myrtanyl amine (1e), confirmed interesting CB1R affinity and selectivity values. Moreover, this study allowed us to highlight that the steric hindarance of a bicyclic aliphatic system as myrtanyl (1e) is responsible for partial agonism. In fact, modeling studies evidenced as this group occupies a region of the space different from that occupied by the monocycloalkyl moieties of 1a-d and this could account for the different agonist/antagonist behaviour of 1e with respect to the other analogues.Therefore, although binding data of the novel compounds do not show significant improvement with respect to derivatives 1a and 1b, essentially this study allowed us to make several considerations. First of all, we can confirm the importance of the substituent on the amide group, particularly when it is an aliphatic ring. Furthermore, the steric hindrance of such substituent can modulate their receptor activity: in fact, when carboxamide groups are aliphatic ring, their activity range from antagonism, in the case of single rings, or partial agonism if a bicyclic ring is supported. Finally, we can also conclude that the introduction of an aryl ring on carboxamide moiety led to compounds endowed with less CBR affinity.
A list of modifications could be find below.
Pyrrolidine 1c has already been described in the following patent By Lohray, Braj Bhushan; Lohray, Vidya Bhushan; Srivastava, Brijesh From U.S. (2011), US 8030345 B2 20111004. The patent should be cited.
Authors’ answer: Done as reference [20] in lines 155 and 439.
Minor corrections:
1) P1 l42: Replace Rimonbant by Rimonaban
Authors’ answer: Done.
2) P2 Scheme 1: include the structure of rimonabantStructure of Imiquimod and resiquimod should be added in Figure 1.
Authors’ answer: The structure of rimonabant has been included in Figure 1.
By contrast, the authors do not understand the reason for including the structures of imiquimod and resiquimod. It would not be consistent with the topic of this manuscript.
3) P3 line 90 and 94: Replace –butyrrolactone by gamma-butyrolactone
Authors’ answer: Done.
4) P3 line 92: Replace BOHt by HOBt
Authors’ answer: Done.
5) P3 line 92: Replace Q-NH2 by R-NH2
Authors’ answer: Done.
6) P3 line 101: Replace amine by amine or hydrazine.
Authors’ answer: Done.
7) P4 line 141: Replace Q by R in the general formula of Table 1
Authors’ answer: Done.
8) P5 Align Cmpd number with Ki Values (From 1d to 1j)
Authors’ answer: Done.
9) P11 line 260 What is the concentration of 1b in the third colum (0.05?)
Authors’ answer: Thanks for your comment. The corrected Figure 6B has been inserted.
10) Numbering of compound is a bit confusing between compound 6 and 14, 7 and 15, 8 and 16, 9 and 17, and 10 and 18 respectively. This should be simplified based of the similarity of the compound. 6 as 6a and 14a-c as 6b-d for example.
Authors’ answer: The numbering of compounds reported in the manuscript ranges from 1 to 7. Authors have not drawn structures with a higher numbering.
11) Cytotoxicity exploration using MTT assay should be added in SI.
Authors’ answer: Thanks for your suggestion, however, we do not consider the possibility to evaluate the cytotoxicity of these compounds.

Round 2
Reviewer 2 Report
Most of the required modification have been realized by the author.